

# Genome-wide comparison of four MRSA clinical isolates from Germany and Hungary

Romen Singh Naorem[1], Jochen Blom[2] and Csaba Fekete[1]

[1] Department of General and Environmental Microbiology, University of Pécs, Pécs, Hungary
[2] Bioinformatics & Systems Biology, Justus-Liebig-Universität Gießen, Gießen, Germany

## ABSTRACT

*Staphylococcus aureus* is a drug-resistant pathogen, capable of colonizing diverse ecological niches and causing a broad spectrum of infections related to a community and healthcare. In this study, we choose four methicillin-resistant *S. aureus* (MRSA) clinical isolates from Germany and Hungary based on our previous polyphasic characterization finding. We assumed that the selected strains have a different genetic background in terms of the presence of resistance and virulence genes, prophages, plasmids, and secondary metabolite biosynthesis genes that may play a crucial role in niche adaptation and pathogenesis. To clarify these assumptions, we performed a comparative genome analysis of these strains and observed many differences in their genomic compositions. The Hungarian isolates (SA H27 and SA H32) with ST22-*SCCmec* type IVa have fewer genes for multiple-drug resistance, virulence, and prophages reported in Germany isolates. Germany isolate, SA G6 acquires aminoglycoside (*ant(6)-Ia and aph(3')-III*) and nucleoside (*sat-4*) resistance genes via phage transduction and may determine its pathogenic potential. The comparative genome study allowed the segregation of isolates of geographical origin and differentiation of the clinical isolates from the commensal isolates. This study suggested that Germany and Hungarian isolates are genetically diverse and showing variation among them due to the gain or loss of mobile genetic elements (MGEs). An interesting finding is the addition of SA G6 genome responsible for the drastic decline of the core/pan-genome ratio curve and causing the pan-genome to open wider. Functional characterizations revealed that *S. aureus* isolates survival are maintained by the amino acids catabolism and favor adaptation to growing in a protein-rich medium. The dispersible and singleton genes content of *S. aureus* genomes allows us to understand the genetic variation among the CC5 and CC22 groups. The strains with the same genetic background were clustered together, which suggests that these strains are highly alike; however, comparative genome analysis exposed that the acquisition of phage elements, and plasmids through the events of MGEs transfer contribute to differences in their phenotypic characters. This comparative genome analysis would improve the knowledge about the pathogenic *S. aureus* strain's characterization, and responsible for clinically important phenotypic differences among the *S. aureus* strains.

Corresponding author
Csaba Fekete,
fekete@gamma.ttk.pte.hu

## INTRODUCTION

*Staphylococcus aureus* is a notorious nosocomial, and community-acquired pathogen (*Chambers & Deleo, 2009*). It has the capability of colonizing diverse ecological niches within its human host, including the skin, blood, respiratory tract, and nasal passages (*Deleo, Diep & Otto, 2009*) and causing diverse ranges of the hospital and community-acquired infections such as skin and soft tissue infections (SSI) for example, carbuncles, abscesses, styes, and impetigo and life-threatening infections such as bacteremia, necrotizing pneumonia, osteomyelitis, endocarditis, and sepsis (*Götz, Bannerman & Schleifer, 2006*; *Mottola et al., 2016*). Methicillin-resistant *S. aureus* (MRSA) acquired a mobile genetic element called Staphylococcal chromosomal cassette *mec* (*SCCmec*) accompanied by methicillin resistance gene (*mecA*) (*Zhang, Mcclure & Conly, 2012*). The β-lactam insensitive protein, penicillin-binding protein (PBP2a) encoded by *mecA* gene reduces affinity to penicillin and β-lactam antibiotics including methicillin, oxacillin, cefoxitin, *etc.,* and develop resistance toward the β-lactam antibiotics (*Jansen et al., 2006*; *Mistry et al., 2016*). MRSA acquires an arsenal of antibiotic resistance genes (ARGs) and virulence factor encoding genes (VFGs) through horizontal gene transfer (HGT) and recombination (*Chan, Beiko & Ragan, 2011*; *Hughes & Friedman, 2005*).

MRSA can anchor and colonize on epithelial surfaces and produce biofilm (*Goudarzi et al., 2018*). The biofilm produced by MRSA strains encase its cells in the exopolysaccharide matrix reduces the activity of antibacterial agents and immune defense. The dispersal of bacterial cells from the biofilm can result in secondary site infections and leading infections worsen and difficult to eradicate (*Lister & Horswill, 2014*). Biofilm formation is a complex process that consists of an extracellular polymeric matrix (ECM) formation involving polysaccharide intercellular adhesin (PIA), protein-protein interactions, and the incorporation of extracellular DNA (eDNA) (*O'Gara, 2007*; *Payne & Boles, 2015*). The biofilm formation is determined by the *icaADBC* gene cluster, responsible for PIA and capsular polysaccharide/adhesion synthesis (*Chaieb, Mahdouani & Bakhrouf, 2005*). MRSA possesses adhesive matrix molecules that are encoded by elastin (*ebps*), laminin (*eno*), clumping factors A and B (*clfA* and *clfB*), fibronectin A and B (*fnbA* and *fnbB*), collagen (*cna*), fibrinogen (*fib*), bone sialoprotein (*bbp*), *etc* (*Lindsay et al., 2006*). These molecules are exported to the bacterial cell surface to enable adherence with host tissues, leading to play a role in pathogenesis (*Mazmanian, 1999*).

*S. aureus* acquires an arsenal of ARGs and VFGs that are subjected to HGT and recombination (*Chan, Beiko & Ragan, 2011*; *Hughes & Friedman, 2005*). Hospital-associated MRSA (HA-MRSA) is often associated with metastatic infections and significant morbidity and mortality (*Gould, 2005*). However, Community-associated MRSA (CA-MRSA) infections have seen a high increase in prevalence, posing a greater threat to the public (*Morens & Fauci, 2013*). The genomic plasticity of *S. aureus* has facilitated the development of hypervirulent and drug-resistant strains, result in challenging issues to antibiotic treatment and health concern.

The classical techniques such as antibiotic susceptibility test (AST) patterns and molecular typing methods such as *SCCmec*-typing, Pulse-Field Gel Electrophoresis (PFGE),

Multi-Locus Sequence Typing (MLST), Multi-locus variable-number tandem-repeat (VNTR) analysis (MLVA), *S. aureus* protein A (*spa*)-typing, accessory gene regulator (*agr*)-typing are widely used to detect and differentiate several MRSA strains, and helpful for identifying the risk factors associated with MRSA infection which support the establishment of adequate infection control programs (*Zhang, Mcclure & Conly, 2012*; *Mistry et al., 2016*). However, these methods are expensive and time-consuming, and have limitations in infection control and investigating the nosocomial transmission due to low resolution (*Du et al., 2011*). In this modern era, whole-genome sequence-based typing offers an excellent resolution in global and local epidemiologic investigations of pathogen outbreaks and offers further data mining activities essentially for ARGs and VFGs profiling (*Köser et al., 2012*). So, the Next Generation Sequencer (NGS) based-genome sequencing technique has become an essential tool in the clinical microbiology arenas for comparative genomic analysis of several other species of the *Staphylococcus* genus in terms of the niche adaptation, combat antibiotics, and emergence of new virulent strains in real-time.

In our preliminary study, the polyphasic characterization of 35 *S. aureus* strains originated from Germany, and Hungary was performed. This characterization included antibiotic resistance test (ART), biochemical tests, biofilm-forming assay, and PCR based typing techniques involving the amplification of *mecA*, *pvl*, *SCCmec*-type, *spa* type, *coa-HaeIII*-RFLP, and biofilm-associated genes. Principal component analysis from polyphasic characterization data showed that the strains originated from the same geographical region were found in the close group while SA G8, Germany strain was grouped with other Hungarian strains (*Naorem et al., 2020*). The Hungarian strains (SA H27 and SA H32) belonged to the same Clonal Complex (ST22/*SCCmec*-IV) were clustered in the same group; however, these strains were isolated from the different sites of infections (nostrils and trachea) and showed different antibiotic resistance patterns and biofilm-forming abilities. Similarly, the strains collected from Germany *viz.,* SA G6, and SA G8 belonged to the same Clonal Complex (ST228/*SCCmec*-I and ST225/ *SCCmec*-II) having similar antibiotic resistance pattern, and biofilm-forming profiles, but these strains were isolated from the different site of infections (skin and other body sites) and not clustered in the same group (*Naorem et al., 2020*). Based on this information, these four *S. aureus* strains were chosen for in-depth comparative genome levels study to better understand the genomic differences among the strains. We assumed that the selected strains have a different genetic background in terms of the presence of ARGs, VFGs, prophages, plasmids, and secondary metabolite biosynthesis genes that may play a crucial role in niche adaptation and pathogenesis. To clarify these assumptions, we performed a comparative genome analysis of these four strains and observed many differences in their genomic compositions.

## MATERIALS & METHODS

### Bacterial strains used in this study

In this study, four *S. aureus* isolates collected from Germany (SA G6, and SA G8) and Hungarian (SA H27, and SA H32) were used. Hungarian isolate, SA H27 was reported as a strong biofilm producer among them (*Naorem et al., 2020*).

## pH tolerance assay

*S. aureus* strains were cultured overnight at 37 °C in tryptic soy broth (TSB) (DB, Germany). The cell density (colony forming units, CFU) was adjusted to a final concentration of $\sim 10^6$ CFU/ml in pH 4.5 TSB and pH 9.5 TSB. Cell suspension (200 µl) were loaded into the 96-well flat-bottomed polystyrene microtiter plate (Costar 3599; Corning; USA). The plates were incubated at 37 °C for 24 h without shaking, then the growth was measured at 492 nm wavelength using a Multiskan Ex microtiter plate reader (Thermo Electron Corporation, USA). The experiments were performed in triplicate and analyzed using GraphPad Prism 6 software package (Graphpad Software Inc, San Diego, CA, USA).

## Genomic DNA isolation and sequencing

The genomic DNA was extracted using the GenElute[TM] Bacterial Genomic DNA Kit (Sigma, USA) following the manufacturer instructions. The concentration and purity of genomic DNA was measured using dsDNA HS (High Sensitivity) Assay Kit in Qubit 3.0 fluorometer (Thermo Fisher Scientific Inc., Waltham, MA, USA) and subsequently DNA quality was visualized by agarose gel electrophoresis.

Genomic libraries were prepared by using the NEB Next Fast DNA Fragmentation and Library Preparation Kit, developed for Ion Torrent (New England Biolabs) and used according to 200 bp protocol. After chemical fragmentation, DNA size selection was performed on precast 2% E-Gel Size Select Gel (Thermo Fisher Scientific Inc., Waltham, MA, USA). The quality of the libraries was verified using Agilent high sensitivity DNA assay kit (Agilent Technologies Inc., Santa Clara, CA, USA) in Agilent 2100 Bioanalyzer System (Agilent Technologies Inc., Santa Clara, CA, USA). For the template preparation, Ion PGM Hi-Q View OT2 Kit was used (Thermo Fisher Scientific Inc., Waltham, MA, USA). The template positive beads were loaded on Ion 316v2 Chip and sequenced using Ion PGM Hi-Q View Sequencing Kit on Ion Torrent PGM sequencer (Thermo Fisher Scientific Inc., Waltham, MA, USA).

## Genome assembly and annotation

In-silico trimming of adapter and barcode sequences and data analysis were performed using Torrent Suite 5.4.0 (Thermo Fisher Scientific Inc., Waltham, MA, USA) and the trimmed paired-end reads were assembled by de novo assembler SPAdes 3.7.1 software with 21, 33, 55, 77, 99, 127 k-mer values (*Nurk et al., 2013*). The assembly-stats and quality of genome completeness for each strain were estimated using the web platform QUEST (*Gurevich et al., 2013*). For identifying the closely related strains, the genome assemblies were analyzed by the kmerFinder 3.1 web platform (*Larsen et al., 2014*). The genome assembly was aligned against the reference genome for the contigs rearrangement using the 'Move Contigs' algorithm in Mauve 2.4.0 (*Darling, Mau & Perna, 2010*) and further, scaffolds were generated with reference genome/ genome of closely related strains predicted by kmerFinder 2.0 as a guide for alignment using the reference-based scaffolder MeDuSa (*Bosi et al., 2015*). Gene annotation of the genome assemblies was performed via the fully automated RAST (Rapid Annotation using Subsystem Technology) (*Aziz et al., 2008*) and PATRIC 3.5.7 (Pathosystems Resource Integration Center) (*Wattam et al., 2013*) pipelines using the reference genome.

## In-silico characterization of genome assemblies

In-silico epidemiologic characterization of genome assemblies was performed using SCCmecFinder-1.2 for the identification of *SCCmec* types (*Kaya et al., 2018*), *spa* Typer 1.0 (*Bartels et al., 2014*) for *spa* type, and MLST 1.8 (*Larsen et al., 2012*) for Multilocus Sequence Type in a web-based server provided by the Center for Genomic Epidemiology (https://cge.cbs.dtu.dk/services/). In-silico *arg* (accessory gene regulator)-typing was performed using the primers described by *Shopsin et al. (2003)* in in-silico PCR amplification tools (*Bikandi et al., 2004*).

The genome assemblies were screened for plasmid replicon (*rep*) genes using PlasmidFinder 2.1 (*Carattoli et al., 2014*) with default parameters. The identified nonaligned contig or scaffold associated with plasmid sequences were extracted and used for the identification of full-length plasmid regions using PLSDB (Plasmid Database) version-2020-03-04 (*Galata et al., 2018*) with search strategy Mash screen, and the default values were a maximum *P*-value of 0.1 and a minimum identity of 0.99 (https://ccb-microbe.cs.uni-saarland.de/plsdb/). Identified plasmids were compared with the closest reference plasmids using Easyfig version 2.2.3 (*Sullivan, Petty & Beatson, 2011*). The identification and annotation of prophage sequences were performed by screening the genome assemblies using PHASTER (PHAge Search Tool Enhanced Release) (*Arndt et al., 2016*), and identified template phages were classified for their lifestyles using PHACTS (Phage Classification Tool Set) (*McNair, Bailey & Edwards, 2012*).

In-silico mining of candidate ARGs and VFGs were performed using CARD (Comprehensive Antibiotic Resistance Database) version 3.0.8 in RGI (Resistance Gene Identifier) version 5.1.0 platform (https://card.mcmaster.ca/analyze/rgi) (*Alcock et al., 2019*), and a comprehensive set of *S. aureus* VFGs was analyzed using VFDB (Virulence Factor Database) in VFanalyzer (*Liu et al., 2018*) and the PATRIC tool version 3.6.3 (https://www.patricbrc.org/) (*Wattam et al., 2013*). Further, heatmap and hierarchical clustering were generated to visualize the presence and absence of VFGS and ARGs in *S. aureus* strains using a web-based application, Morpheus, (https://software.broadinstitute.org/morpheus). Secondary metabolite biosynthesis gene clusters and the detection of genes encoding bacteriocins were analyzed using antiSMASH 5.0 (*Blin et al., 2019*) and BAGEL4 (*Van Heel et al., 2018*). The prediction of chromosomal genomic islands was predicted by using IslandViewer 4 (*Bertelli et al., 2017*).

## Comparative genomic analysis

The ANI was determined based on BLAST+ using the JSpeciesWS webserver (*Richter et al., 2015*). The pairwise comparisons between the genomes of *S. aureus* isolates and their nearest reference genomes were conducted using GBDP (Genome BLAST Distance Phylogeny) under the algorithm trimming and distance formula d5, and calculated each distance with 100 replicates (*Meier-Kolthoff et al., 2013*). dDDH (Digital DNA-DNA Hybridization) values and confidence intervals were calculated using the recommended settings of the GGDC 2.1 (*Meier-Kolthoff et al., 2013*).

Genomes of *S. aureus* isolates and their reference strains were compared with CGViewer (Circular Genome Viewer) server (*Grant & Stothard, 2008*). The functional annotation was

performed using EggNOG (Evolutionary Genealogy of Genes: Non-supervised Orthologous Groups) mapper 5.0 database (*Huerta-Cepas et al., 2018*) and RAST server-based SEED viewer (*Overbeek et al., 2013*).

The pan-genome, core-genome, and singletons were calculated using four study genomes of *S. aureus* isolates in EDGAR version 2.0 software framework (*Blom et al., 2016*). This pan-genome analysis was extended using four study genomes coupled with three reference genomes of *S. aureus* strains. The core-genome was analyzed in the genomes set using reciprocal best BLAST hits of all CDS using EDGAR version 2.0 software framework (*Blom et al., 2016*). The singletons were calculated for the contig of a strain by comparing to the CDS of a set of contigs in EDGAR. The CDS that has no match with SRV (Score Ratio Value Plots) higher or equal the master cut-off in any of the contigs were considered as singletons. The development of pan-genome and core-genome sizes was analyzed using the core/pan development feature and as well, the pan *vs.* core development plot was generated in EDGAR. Heap's Law function was applied to calculate whether the pan-genome open or closed using the equation $n = k * N^{(-\alpha)}$ where n = expected a number of genes; N = number of genomes; $k$ and α (α = 1 − γ) are proportionality constant and exponent, respectively (*Tettelin et al., 2008*). Heap's law predicted that closed pan-genome (when α > 1 (γ < 0)), and open pan-genome (when α < 1 (0 < γ < 1)). According to *Tettelin et al. (2008)*, core-genome and singletons developments were calculated by the least-square fitting of exponential decay functions.

The Rcp (ratio of core-genome to that of pan-genome) was calculated (*Ghatak et al., 2016*). Then, genomic subsets, including the number of core-genome and singletons in the gene pool, were extracted, and flowerplot was drawn using *in-house* R scripts.

### Phylogenetic analysis

The genome assemblies of the isolates were used for a whole genome-based phylogeny analysis using TYGS (Type/Strain Genome Server) (*Meier-Kolthoff & Göker, 2019*) engaging with genomes of closely related strains of *S. aureus*. The phylogenomic trees were reconstructed using FastME 2.1.6.1 (*Lefort, Desper & Gascuel, 2015*) from the GBDP (Genome BLAST Distance Phylogeny) distances calculated from genome sequences under the algorithm 'coverage' and distance formula d5 (*Meier-Kolthoff et al., 2013*). The trees were rooted at the midpoint (*Farris, 1972*); branch supports were inferred from 100 pseudo-bootstrap replicates and visualized with Interative Tool Of Life v4 (iTOL) (*Letunic & Bork, 2019*). The core SNPs of genome sequences were extracted using Panseq (*Laing et al., 2010*) and the phylogenetic tree was constructed using PhyML+SMS module in NGPhylogeny.fr (*Lemoine et al., 2019*) to select the best evolutionary model, further the tree was annotated in Interative Tool Of Life v4 (iTOL) (*Letunic & Bork, 2019*).

## RESULTS

The *S. aureus* isolates could survive at pH 4.5 through pH 9.5 conditions with a survival rate of ∼45%–84%. SA G8 isolate showed the highest cell survival rate of 84.4% at acidic pH but its cell survival rate drops down by 7% when subjected to alkaline pH conditions (Table S1).

## General genomic features of *S. aureus* isolates

The genomic DNA of *S. aureus* isolates was successfully sequenced in the IonTorrent PGM sequencing platform. The average raw reads obtained from the genome sequencing of SA G6, SA G8, SA H27, and SA H32 are ∼88.9, 69.6, 128.3, and 92.7 million bases (Mb) for genomes of SA G6, SA G8, SA H27, and SA H32 strains respectively. The closely related strains identified by kmerFinder 2.0 were *S. aureus subsp. aureus* ST228 (HE579073), *S. aureus subsp. aureus* JH9 (CP000703) for SA G6 and SA G8 strains, respectively. Also, *S. aureus subsp. aureus* HO 5096 0412 (HE681097.1) was identified closely related strains for SA H27 and SA H32 strains. Among the *S. aureus* isolates, SA G8 has the largest genome size (28633393 bp) with high % GC content (32.81%). The numbers of protein-coding sequences (CDSs) in the *S. aureus* strains varied from 2630 (SA H27) to 2743 (SA G8). The comparison of draft genome assemblies, genome annotation, molecular typing, plasmid, and prophage features for *S. aureus* genomes were summarized in Table 1.

## Genes encoding plasmids

The putative plasmids were detected in nonaligned contigs or scaffolds that exhibited an unexpected high coverage level after the genome assemblies. A putative plasmid (p1G6) of 13331bp length was identified at Scaffold 4 of the SA G6 genome consisting of the replication gene (*repA*). The p1G6 plasmid has 30.97% sequence coverage with plasmids pTW20_1 (FN433597.1) (Fig. 1A). The sequence coverage region of p1G6 with pTW20_1 constitutes the genes that encode for proteins such as IS6 family transposase, replication-associated protein (Rep), cadmium resistance transporter (CadD), cadmium efflux system accessory protein (CadX), replication initiation protein A (RepA), quaternary ammonium compound efflux MFS transporter (QacA), multidrug-binding transcriptional regulator (QacR), DUF536 domain-containing protein (mP), AAA family ATPase (Abp), hypothetical proteins, HAD hydrolase family protein, and IS257 family transposase. The SA H32 genome also consists of a putative plasmid (p2H32) having a length of 2530 bp located at Scaffold 3 and showed 71.32% sequence coverage with plasmids AR_0472 (NZ_CP029648.1). It consists of a replication gene (*repL*) and carried an erythromycin resistance gene (*emrC*) (Fig. 1B). The identified plasmids of *S. aureus* encode no other factors for their transfer, such plasmids may transfer via phage transduction (*McCarthy & Lindsay, 2012*). The linear graphical map of plasmid comparison was represented in Fig. 1.

## Characteristic of prophages-like elements

The genomes of *S. aureus* isolates have several prophages and phage-like element regions and these prophages were belonged to the *Siphoviridae* family and having temperate lifestyles. The highest number of prophage regions was found in the genome of SA G8 isolate including three intact prophages (phiG8.2, phiG8.3, and phiG8.4), a questionable (phiG8.1), and an incomplete (phiG8.5) prophages. Four prophage regions were found in the genome of SA G6 isolate including an intact prophage (phiG6.3), two questionable prophages (phiG6.1 and phiG6.4), and an incomplete prophage (phiG6.2). The genome of SA H27 isolate harbor three intact prophages (phiH27.1, phiH27.2, and phiH27.3) while

**Table 1** General genomic features of *S. aureus* genomes in this study.

| Strains | SA G6 | SA G8 | SA H27 | SA H32 |
|---|---|---|---|---|
| Size (bp) | 2856214 | 2857863 | 2783185 | 2786627 |
| Contigs | 103 | 83 | 44 | 63 |
| Scaffolds | 22 | 15 | 1 | 3 |
| N50 (bp) | 125160 | 263953 | 328241 | 208577 |
| GC% | 32.79 | 32.81 | 32.73 | 32.72 |
| CDS | 2734 | 2743 | 2630 | 2657 |
| Genes assigned to SEED | 2101 | 2169 | 2014 | 2036 |
| rRNA | 9 | 10 | 8 | 9 |
| tRNA | 61 | 60 | 57 | 60 |
| Prophage Regions | 3 | 5 | 3 | 1 |
| [a]Plasmids | p1G6 | - | - | p2H32 |
| *SCCmec* type | I | II | IVa | IVa |
| MLST | ST228 | ST225 | ST22 | ST22 |
| *Spa* type | t535 | t003 | t379 | t1258 |
| *agr*-type | II | II | I | I |
| Accession no. | RAHA00000000 | QZFC00000000 | CP032161 | RAHP00000000 |

**Notes.**
[a]Plasmids: The presence of plasmid in genome is indicated by plasmid name, while absent is represented by a minus (-) sign.

the genome of SA H32 harbor only one intact prophage (phiH32.1). The *lukF-PV* and *lukM* genes (Bicomponent leukotoxins), and *plc* gene (Phospholipase C) were identified in the prophages of phiG6.4, phiG8.4, phiH27.2, and phiH32.1. The prophages of phiG6.3, phiG8.4, and phiH27.2 carried *sak* gene (staphylokinase) and *scn* gene (staphylococcal complement inhibitor). Chemotaxis inhibitory protein encoded by *chp* gene was identified in phiG8.4 and phiH27.2 prophages. Enterotoxin A encoded by *sea* gene was harbored by the prophages of phiG6.3 and phiG8.4. Hemolysin genes such as *hlb* ( β-hemolysin), and *hlgB* (-hemolysin B) were found in the prophages of phiH27.2, and phiH32.1. In addition to virulence factors, phiG6.4 prophage carried ARGs genes that conferred resistance to beta-lactamase (*blaZ*), aminoglycoside (*ant(6)-Ia* and *aph(3′)-III*) and nucleoside (*sat-4*) antibiotics. The comparative analysis of VFGs associated with putative prophages was summarized in Table S2.

## In-silico analysis of antimicrobial resistance and associated genes in the genomes

Four study genomes of *S. aureus* isolates shared 63.3% (19/30) of antibiotic resistance and associated genes (Fig. 2). The shared genes comprise methicillin-resistant PBP2a (*mecA* and *mecR1*); multidrug resistance efflux (*ygaD*); fluoroquinolone (*norA* and *gyrA*); fluoroquinolone and acridine dye (*arlS* and *arlR*); glycylcycline (*mepA*); tetracycline (*tet-38*); tetracycline, penam, cephalosporin, glycylcycline, rifamycin, phenicol, triclosan, fluoroquinolone (*mgrA* and *marR*); lipopeptide (*pgsA*, *clsA* and *rpoC*); rifampicin (*rpoB*); aminocoumarin (*gyrB* and *parE*); dihydrofolate reductase (*dfrA/folA*) and defensin (*mprF/fmtC*, multiple peptide resistance factor) that play roles in resistance mechanism including antibiotic efflux, antibiotic target alteration, and antibiotic target replacement.

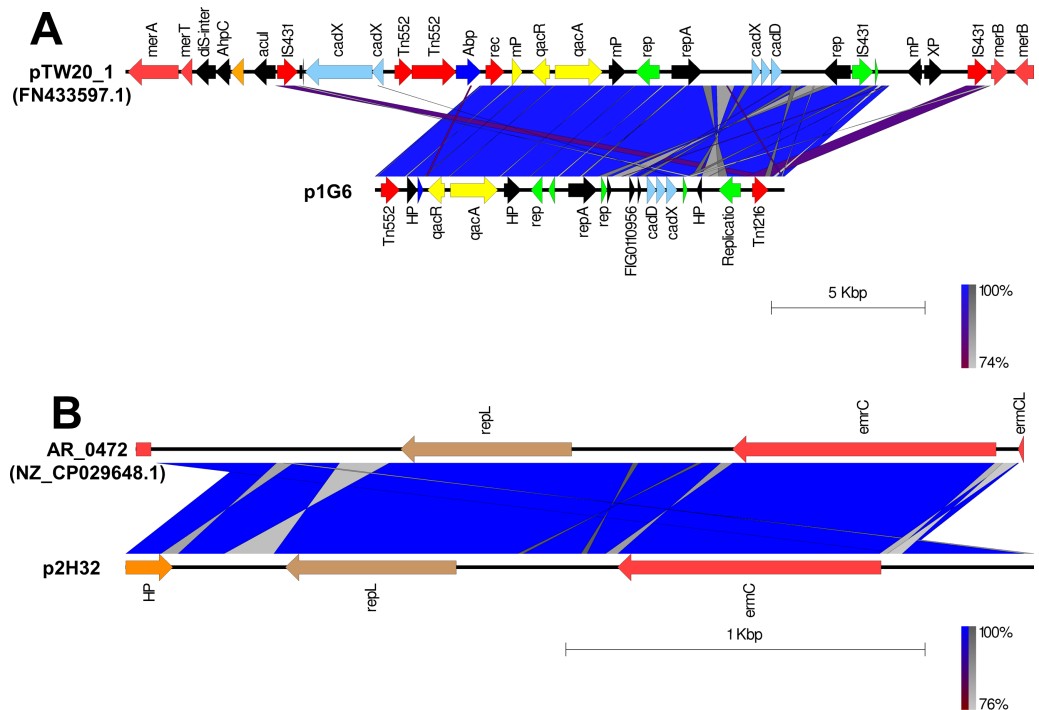

**Figure 1 Comparison of linear plasmid maps by Easyfig alignment.** Coding Sequences are represented by colored arrows. Blue lines between the plasmids indicate the shared similarity regions according to BLASTn identity. CDS are characterized by functions as follows: antiseptic resistance genes (yellow), erythromycin resistance gene (light red), DNA replication (green), transposons/integrases (red), replication A gene/ hypothetical proteins/others (black), replication L gene (brown), and cadmium resistance gene (cyan). The outer scale is marked in kilobases. (A) Sequence alignment of p1G6 plasmid of SA G6 isolates with the reference pTW20_1 (FN433597.1) plasmid. (B) Sequence alignment of p2H32 plasmid of SA H32 with the reference AR_0472 (NZ_CP029648.1) plasmid.

The comparative analysis of ARGs revealed that the genome of SA G6 isolate acquired additional ARGs responsible for the resistance of aminoglycoside (*aph (3′)-IIIa, ant (6′)-I* and *aac (6′)-II*), nucleoside (*sat*), fluoroquinolone (*qacA*). The macrolide, lincosamide, streptogramin (MLS) erythromycin antibiotic resistance genes (*emrA*) were detected in the genomes of SA G6 and SA G8 isolates while the genome of SA H32 isolate present *emrC* gene. The penicillin resistance gene (*blaZ*) was found absent in SA G8 isolate. This in-silico identification and our previous antibiotic susceptibility test results were correlated with beta-lactam, erythromycin (MLS), and vancomycin antibiotic resistance analysis (*Naorem et al., 2020*). The secondary metabolite biosynthetic gene clusters identified among the genomes were staphylobactin, aureusimine, bacteriocin, and staphyloferrin A. The auto-inducing peptide (AIP)-II gene was identified in SA G6 and SA G8 genomes while AIP-I gene was identified in SA H27 and SA H32 genomes.

### In-silico analysis of virulence-factors encoding genes in the genomes

The VFGs predicated against the VFDB revealed 59 VFGs were shared in all strains that are responsible for adherence, toxin, anti-phagocytosis immune evasion, secretion system,

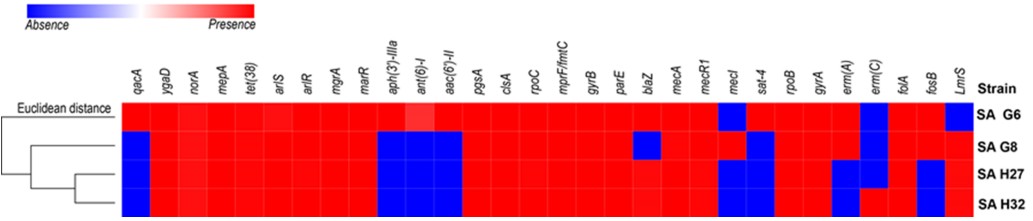

**Figure 2** **Heat map showing the presence (red color) and absence (blue color) of antibiotic resistance genes.** The labels on top indicate the gene names and the label on the left indicates the strains.

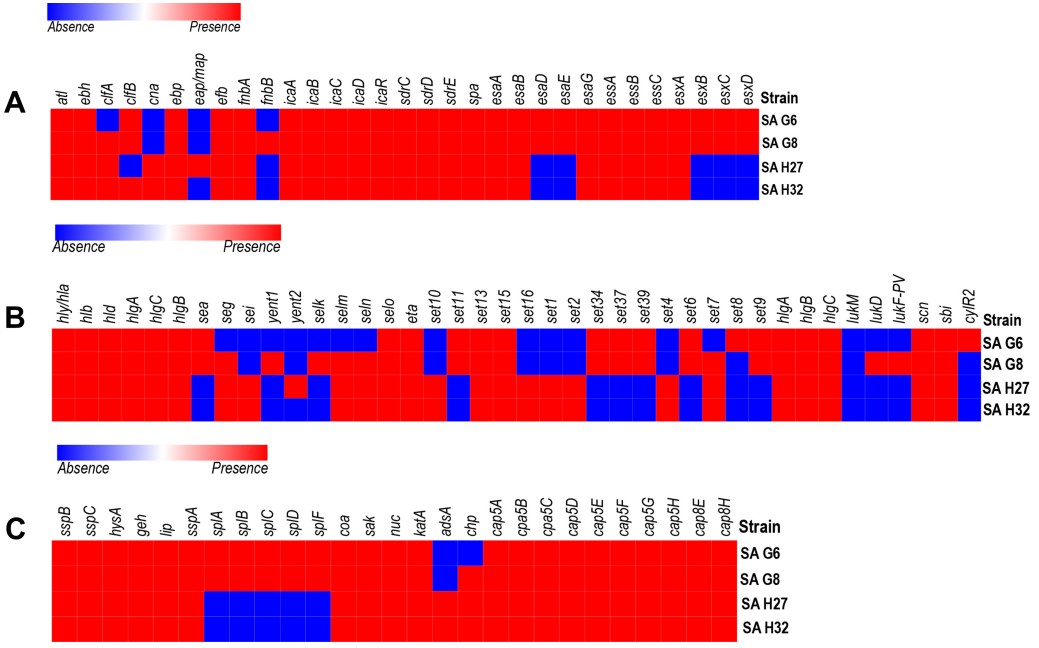

**Figure 3** **Heat map showing the presence (red color) and absence (blue color) of virulence factors encoding genes.** (A) Adherence and secretary factors. (B) Toxins. (C) Enzymes and anti-phagocytosis (capsules) factors. The labels on top indicate the gene names and the label on the left indicates the strains.

exoenzyme activity, and iron uptake (Fig. 3). The genome of SA G8 isolate has occupied 3.40% of VFGs against its CDS, whereas the genome of SA H32 isolate has 2.97% of VFGs against its CDS.

Adherence associated genes shared in all genomes of *S. aureus* isolates were 63.63% (14/22) such as autolysin (*atl*), cell wall-associated fibronectin-binding protein (*ebh*), elastin binding protein (*ebp*), fibrinogen binding protein (*efb*), fibronectin-binding proteins A (*fnbA*), intercellular adhesin (*icaA*, *icaB*, *icaC*, *icaD*, *icaR*), ser-Asp rich fibrinogen-binding proteins (*sdrC*, *sdrD*, sdrE), staphylococcal protein A (*spa*) (Fig. 3A). The genome of SA G8 isolates present 77.27% (17/22) of adherence associated genes with additional genes of clumping factor A (*clfA*), clumping factor B (*clfB*), and fibronectin-binding proteins (*fnbB*).

Type VII secretion system involves in membrane-associated proteins (*esaA, essA, essB, and essC*), soluble cytosolic (*esaB, esaE, esaG*), and secreted virulence factors (*esxA, esxB, esxC, esxD,* and *esaD*) were identified in the genomes of SAG6 and SA G8 isolates while the SAH27 and SAH32 isolates absence *esaD, esaE, esxB, esxC,* and *esxD* genes (Fig. 3A).

The genomes of all isolates shared 29.41% (10/34) of toxin genes such as alpha-hemolysin gene (*hla*), beta-hemolysin gene (*hlb*), delta hemolysin gene (*hld*), gamma hemolysin A (*hlgA*), gamma hemolysin B (*hlgB*), gamma hemolysin C (*hlgC*), enterotoxin-like O (*selo*), exfoliative toxin type A (*eta*), and exotoxin (*set13, set15*) (Fig. 3B). The highest number of toxin genes were identified in the genome of SA G8 *i.e.,* 73.52% (25/34), and in addition to shared genes, the extra genes were enterotoxin A (*sea*), enterotoxin B (*seg*), enterotoxin Yent1 (*yent1*), enterotoxin-like K (*selk*), enterotoxin-like M (*selm*), enterotoxin-like N (*seln*), exotoxin (*set6, set7, set9, set11, set34, set37, set39*), leukotoxin D (*lukD*), Panton-Valentine leukocidin (*lukF-PV*).

The genes involve in anti-phagocytosis namely capsular polysaccharide synthesis genes belong to stereotype 5 and 8 predominantly present in all the genomes of isolates were capsular polysaccharide synthesis enzyme Cap5A (*cap8A*), capsular polysaccharide synthesis enzyme Cap5B (*cap8B*), capsular polysaccharide synthesis enzyme Cap5C (cap8C), probable polysaccharide biosynthesis protein EpsC (*cap8D*), capsular polysaccharide synthesis enzyme Cap8E (*cap8E*), capsular polysaccharide synthesis enzyme Cap5F (*cap8F*), UDP-N-acetyl-L-fucosamine synthase (*cap8G*), capsular polysaccharide synthesis enzyme Cap5L (*cap8L*), capsular polysaccharide synthesis enzyme Cap8M (*cap8M*), capsular polysaccharide synthesis enzyme Cap8N (*cap8N*), UDP-N-acetyl-D-mannosamine dehydrogenase (*cap8O*) and UDP-N-acetylglucosamine 2-epimerase (*cap8P*) (Fig. 3C). Other genes responsible for the host immune evasion such as IgG-binding protein (*sbi*), staphylococcal complement inhibitor (*scn*), and chemotaxis inhibiting protein (*chp*) were identified in all isolates.

Several exoenzymes encoding genes namely cysteine protease/ staphopain (*sspB, sspC*), hyaluronate lyase (*hysA*), lipase (*geh, lip*) serine V8 protease (*sspa*), staphylocoagulase (*coa*), staphylokinase (*sak*), and thermonuclease (*nuc*) were present in the genomes of all isolates. However, five genes cluster for serine protease (*splA, splB, splC, splD, splF*) were absent in the genomes of SA H27 and SA H32 isolates (Fig. 3C).

Eight genes involved in iron uptake mechanism including cell surface protein (*isdA*), cell surface receptor (*isdB*) and cell wall anchor proteins (*isdC*), heme transporter component (*isdD*), high-affinity heme uptake system protein (*isdE*), heme-iron transport system permease protein (*isdF*), sortase B (*srtB*), heme-degrading monooxygenase; staphylobilin-producing (*isdG*) were identified in all the genomes of isolates.

## Comparative genome analysis

The genome comparative analysis based on ANIb matrices results indicated that the genome of SA G8 isolate exhibits the nearest identities to all genomes. SA G8 genome showed ~99.5% identities to SA G6 genome. The genomes of SA G6 and SA G8 exhibited ~99.7% identities to *S. aureus subsp. aureus* ST228 (HE579071.1). Also, the genomes of SA H27 and SA H32 showed 99.9% identities to each other and these two genomes displayed
the highest identities (99.9%) to *S. aureus subsp. aureus* HO 5096 0412 (HE681097.1) (Fig. S1). The digital DDH values between the genome of *S. aureus* isolates and the closest relative genomes were 90.7–100% (using GBDP distance formula d0), 77.1–97.0% (using GBDP distance formula d4), and 91.1–99.9% (using GBDP distance formula d6). SA H27 and SA H32 genomes exhibit the nearest identities of 99.9%, 99.8%, and 100% using the formula d0, d4, and d6 respectively and displayed G+C difference of 0.01%. However, the SA G6 genome showed less identity to all the comparative genomes based on dDDH. The high G+C constituent difference (0.1%) was observed in the case of isolated strains of SA G8 and SA H32 genomes.

A whole-genome circular comparative map of four *S. aureus* genomes and their close reference genomes was generated against *S. aureus subsp. aureus* HO 5096 0412 (HE681097.1) genome using CGView server based on BLAST sequence similarities. Each genome was indicated by a different color, and the darker areas in the circular genome showed a 100% sequence similarity with the reference genome, while the lighter (gray) areas showed a 70% sequence similarity (Fig. 4). The map revealed less gap between the SA H27 (CP032161) and SA H32 (RAHP00000000) genomes showing high proximity between them when compared to other genomes. SA G6 (RAHA00000000) genome has many gaps with white color than the other genomes showing a distant relationship.

The SEED subsystem categories identified by RAST revealed that the genomes of all the isolates possessed "amino acids and derivatives" was the largest subsystem, followed by "carbohydrates", "protein metabolism", and "cofactors, vitamins, prosthetic groups, pigments" (Fig. 5A). The "carbohydrate", and "protein metabolism" subsystems were found largest in SA H27 (12.76%) and SA H32 (10.47%) genomes, respectively. The subsystem belongs to "phages, prophages, pathogenicity island" (2.5%) was identified as highest in the SA G8 genome. Amongst the genomes, the SA G6 genome has the largest subsystem of "amino acids and derivatives" (15.9%), and "virulence, disease, and defense" (4.43%). In the "virulence, disease, and defense" subsystem of SA G6, 93 genes associated with adhesion, bacitracin stress response, colicin v, and bacteriocin production cluster, copper homeostasis, bile hydrolysis, cobalt-zinc-cadmium resistances, multidrug resistances, 2-protein, mercuric reductase, mercury resistance operon, streptothricin resistance, teicoplanin-resistances, aminoglycoside adenylyltransferases, fluoroquinolone resistances, arsenic resistance, fosfomycin resistance, beta-lactamase, cadmium resistance, multidrug resistance efflux pumps, and invasion and intracellular resistances. In the comparative eggNOG function study of *S. aureus* genomes, "amino acid transport and metabolism" was observed as for the majority of COGs, followed by those COGs related to "translation, ribosomal structure, and biogenesis", "transcription", and "cell wall/membrane/envelope". The eggNOG analysis results revealed that the SA G6 genome has the highest number of COGs associated with defense mechanisms (Fig. 5B). In the core genomes, 9.46%, 7.21%, and 6.9% of COGs had functions associated to "amino acid transport and metabolism (E)", "translation, ribosomal structure, and biogenesis (J)", and "transcription (K)", respectively. Amongst the functional prediction of genomes, most COGs were associated with "function unknown (S)".
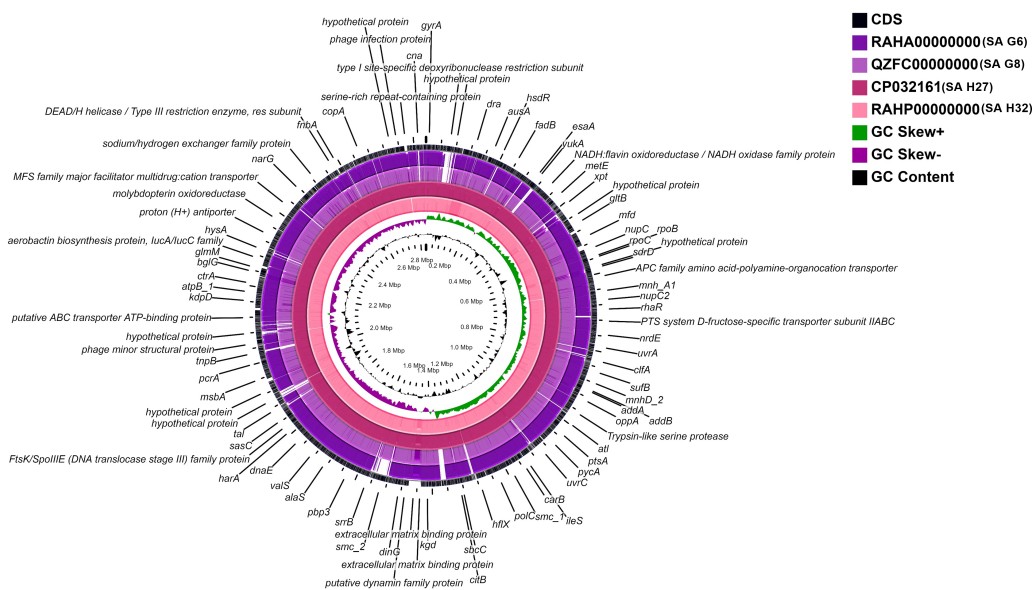

**Figure 4 Circular genome comparison map showing homologous chromosome segment of four *S. aureus* genomes with the reference genome of *S. aureus subsp. aureus HO 50960412* (HE681097.1) strain using CGviewer.** The inner scales designate the coordinates in kilobase pairs (kbp). White spaces indicate regions with no identity to the reference genome.

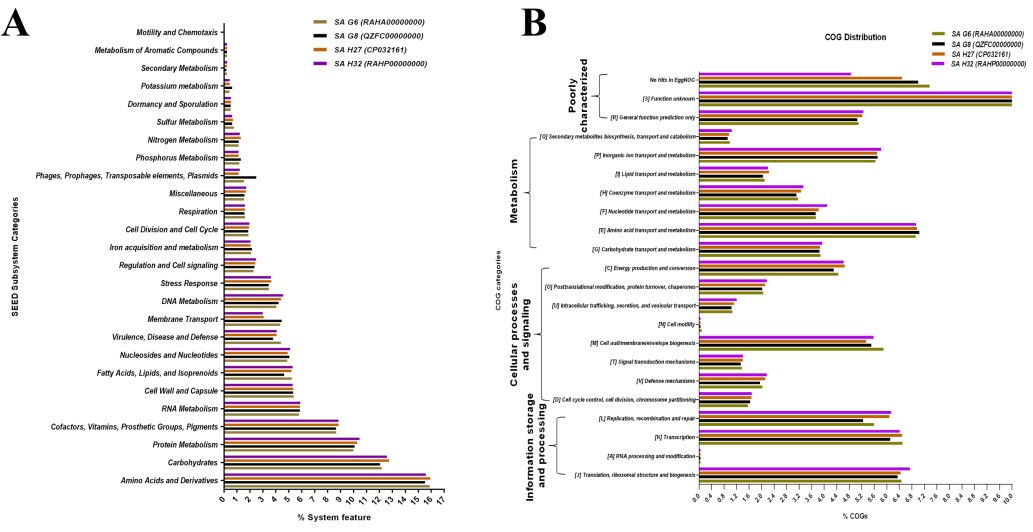

**Figure 5 Comparative functional categorization of all predicted ORFs in the genomes of the *S. aureus* isolates.** (A) Percentage distribution of subsystem categories based on the SEED database. (B) Percentage distribution of COGs based on EggNOG.

## Pan-genome, core-genome, and singletons analysis

The orthologous groups are categories into three groups based on the pan-genome distribution such as core (present in all genomes of *S. aureus* strains), dispensable (present in at least two strains, but not all), and singleton genes (present no orthologs

in any other genomes). The comparison of four study *S. aureus* genomes generated a pan-genome size of 3265 genes, of which 2304 (70.6%) genes were core genome, 462 (14.2%) genes were dispensable, and 499 (15.3%) genes were singletons. The Rcp value for the genomes of *S. aureus* isolates was calculated and the ratio Rcp was 0.70 and it is indicated that the genomes of *S. aureus* isolates were high inter-species diversity. A total of 499 singleton genes were calculated across the genomes of four *S. aureus* isolates, of which SA G6 genome acquired the highest number of singleton genes (220) that constitute the genes encode for proteins *viz.* aminoglycoside 3-phosphotransferase, aminoglycoside 6-phosphotransferase, aminoglycoside N(6)-acetyltransferase, streptothricin acetyltransferase, antiseptic resistance protein, cadmium resistance proteins, cadmium efflux system accessory protein, cadmium-transporting ATPase, ferric siderophore transport system, mercuric ion reductase, anti-adhesin, Tn552 transposase, pathogenicity islands (SaPI and SaPIn2), prophage-like elements, mobile elements, phage associated hypothetical proteins, hypothetical proteins, *etc.* The identified singleton genes of SA G6 genome were present within the genomic island (GI). This GI region is located between 2804353–2873411 base pair sequence region of the genomic sequence. While the SA H27 genome has the least singleton genes (6) constituting the genes encode for hypothetical proteins and phage proteins. The difference in the genomic constituents between the genome of SA H27 and SA H32 isolates revealed that SA H32 acquired the genes encoding for 23S rRNA (adenine(2058)-N(6))-dimethyltransferase, replication and maintenance protein, hypothetical proteins, phage-like elements, and mobile element protein. The genes shared by four study genomes and their respective singletons is represented in Fig. 6A. When the three reference *S. aureus* genomes were included in the pan-genome analysis, the core/pan-genome ratio drop down by 18.97% with inflation of pan-genome to 3415 genes and deflation of core-genome to 1762 genes. The core-genome and singleton genes formed by seven genomes of *S. aureus* strains is represented in flower-plot (Fig. 6B). When the three reference genomes of *S. aureus* strains were included in the pan-genome analysis, SA G6 isolate occupied the highest number of singleton genes (104) while SA H27 isolate has the lowest singleton gene (2) (Fig. 6B). In the pan-genome development analysis of seven *S. aureus* strains, $\alpha$ value (the power-law co-efficient) was estimated as 0.141 which corresponds to the growing and open pan-genome model (Fig. S2). The pan *vs.* core development plot appeared the progression of the pan and core-genomes as additional genomes are added for analysis, and showing that the sharp decline of the core-genome size with the introduction of *S. aureus subsp. aureus* ST228 (HE579073) (Fig. 6C). In the plot of core-genome development, the core-genome size approach ($\Omega$) value revealed that the core-genome size of seven *S. aureus* would be declined to 1404.8 (Fig. S3). The singleton development analysis suggested that the pan-genome size will continue to expand at the rate of 35.9 genes per novel, representative genome (Fig. S4). The shape of the pan-genome *vs.* core-genome curve showed fluctuation in their gene numbers when different order of the genomes was set, even so, pan-genome and core genome developmental plots result remained unaffected by the genomes order.

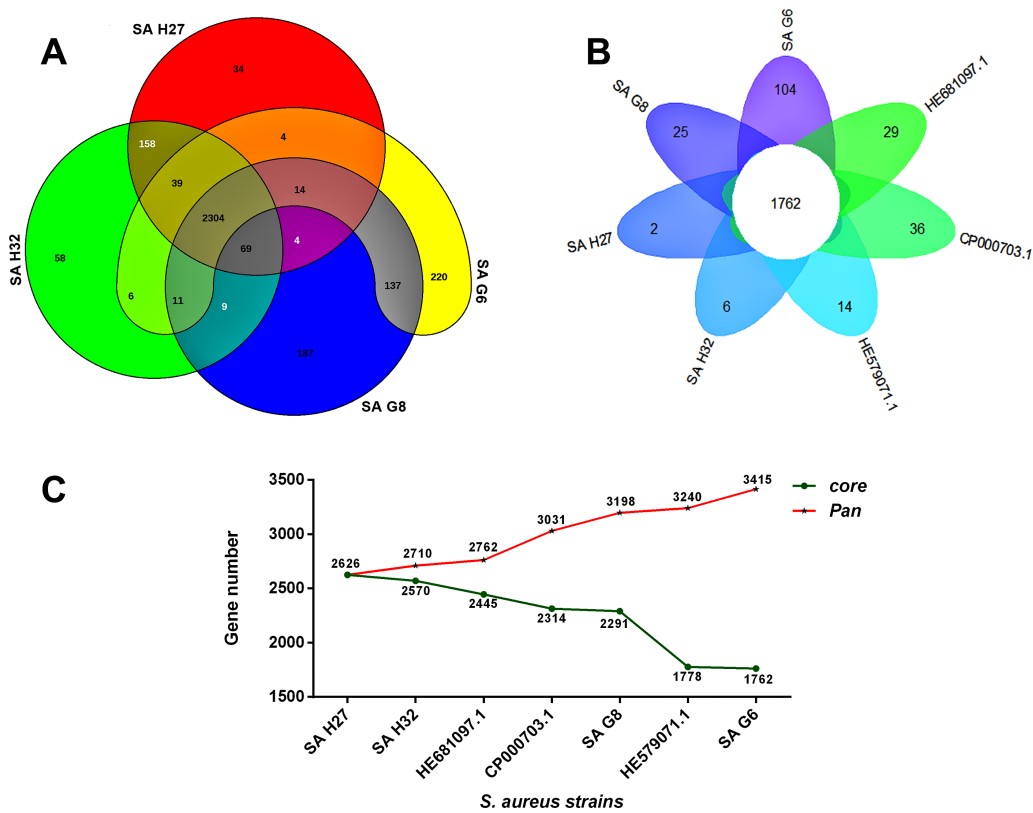

**Figure 6** **Pan-genome analysis of *S. aureus* strains.** (A) The Venn diagram represents the pan and core-genomes of four study genomes (SA G6, SA G8, and SA H27 and SA H32) based on orthology analysis. Overlapping regions represent common CDSs shared between the genomes. The numbers outside the overlapping regions signify the singletons of each genome. (B) Flower plot diagram representing the four study isolates and three reference strains. The core-genes of 1762 was represented in the center of the flower and the petals represent the singletons of concern genomes. (C) Core vs. pan-genome plot of the seven genomes.

## Comparative phylogenetic tree analysis

The phylogenomic analysis of *S. aureus* isolates provide the tree into three major clades (Fig. 7A). The clade A consists of 5 strains that belonged to CC22 and showing that Hungarian isolates, SA H27 and SA H32 have the highest proximity. Germany isolates, SA G6 and SA G8 isolate and other strains belonged to CC5 were clustered in the clade C, showing that SA G6 isolate has closely relatedness to *S. aureus subsp. aureus* ST228 (HE579071.1) and the SA G8 isolate has a higher relatedness to *S. aureus subsp. aureus* JH9 (CP000703.1) than SA G6 isolate.

The phylogenetic relationship inferred from core-genome SNPs holds a similar agreement with the whole genome-based phylogenetic analysis, and these methods could be useful in distinguishing the genomes even in the strain level and phylogenetic trees are illustrated in Fig. 7B.

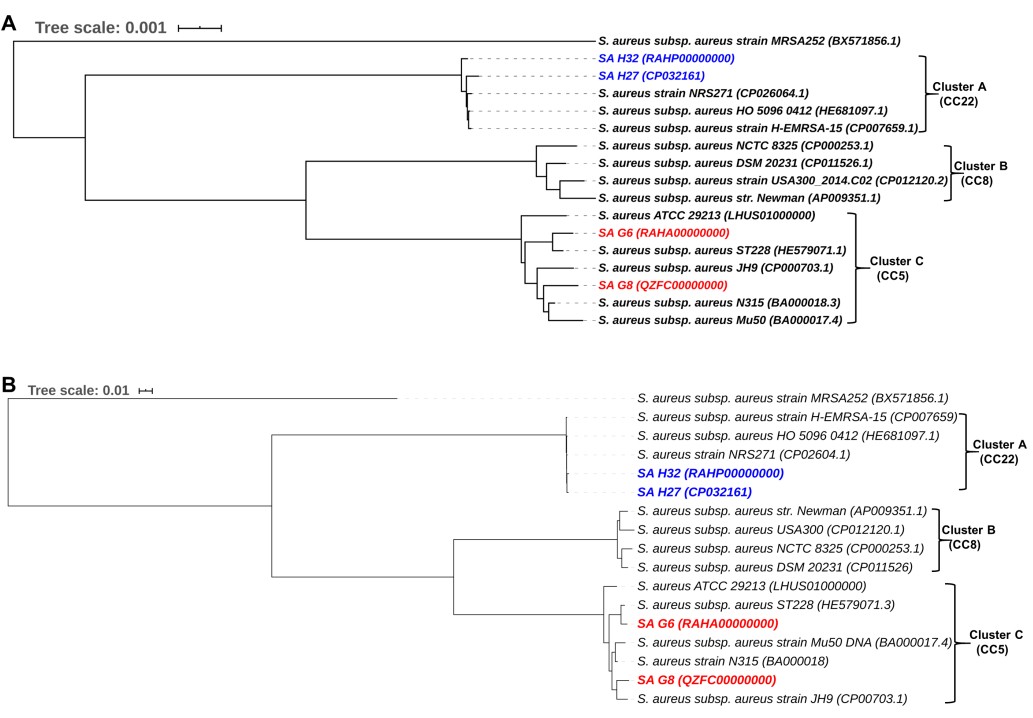

**Figure 7** **Comparative phylogenetic analysis of *S. aureus* isolates strains with their closely related *S. aureus* strains.** Phylogenomic tree generated using closely related genome sequences. The branch lengths are scaled in terms of GBDP distance formula d5 (A), and core-genome SNP tree generated using the alignment of the high-quality SNPs and PhyML+SMS module was applied (B).

## DISCUSSION

*S. aureus* is a significant causative agent of both hospital and community-associated infections (*Chambers & Deleo, 2009*). The study of such pathogen at a molecular level through genome comparative analysis improve the ideas of pathogenesis and evolution. Further, such a study provides advantages in diagnosis, treatments, and infection controls (*Kwong et al., 2015*). In the present study, we used whole-genome sequencing (WGS) and in-silico analysis to determine the comparative ARGs, VRGs, pH tolerance associated genes, and evolutionary relationship of four *S. aureus* isolated from different sites of human infection such as skin, nostril, trachea, and others.

The molecular epidemiology study of MRSA helps to find the risk factors associated with MRSA infections and able to differentiate the several MRSA strains (*Mistry et al., 2016*). The genome-based molecular epidemiology studies found that Germany isolates exhibit *SCCmec* type I with ST228, and *SCCmec* type II with ST225 while Hungarian isolates hold *SCCmec* type IVa with ST22. Also, *agr* type II and I were owned by Germany and Hungarian isolates, respectively (Table 1). According to previous studies suggested that MRSA strains with *SCCmec* types I or II or III are dominant among the HA-MRSA, while *SCCmec* types IV or V are the characteristic of CA-MRSA (*Monecke et al., 2011*; *Chua et al., 2014*). The STs of Germany isolates belonged to CC5 which is typical of HA-MRSA, while Hungarian isolates suggest its relationship CA-MRSA. In hospitals, the multidrug-resistance *SCCmec*

type III was replaced by the multidrug-susceptible *SCCmec* type IV (ST22) strains slowly (*D'Souza et al., 2010*). The Hungarian isolates were found positive to Panton-Valentine Leukocidin (PVL) toxin, which is commonly used as a marker of CA-MRSA (*Shukla et al., 2012*; *Bhutia et al., 2015*) besides this toxin has shown to play a role necrosis, accelerating apoptosis and polynuclear—and mononuclear cells lysis, thereby contributing morbidity and mortality (*Barrera-Rivas et al., 2017*; *Lina et al., 1999*).

Staphylococcal β-lactamase encoded by *blaZ* gene is carried by the transposon Tn*552* or Tn*552*-like elements located on a large plasmid and can be non-inducible or inducible with antibiotics (*Maddux, 1991*). It was noticed that *blaZ* gene was absent in SA G8 isolate, probably due to the curing of *blaZ* positive plasmid (*Pugazhendhi et al., 2020*). Erythromycin resistance gene (*ermA*) was detected in the chromosome of SA G6, and SA G8 isolates, however, *emrC* gene was found in the plasmid of SA H32 (Fig. 1B). It was suggested that these genes may not be involved in the loss of specific ARGs for environmental adaptation, but it is expected to be essential for these isolates (*Lim et al., 2015*).

MRSA is responsible for causing biofilm infections that are more difficult to treat and need more intensive care as compared to *Staphylococcus epidermidis* biofilm (*Reffuveille et al., 2017*). The principal component of biofilm formation is PIA which consists of different intracellular adhesion (*ica*) genes (*Cramton et al., 1999*) and play a crucial role in the initial stage of bacterial cell adherence to surfaces and intercellular adhesion for the cells to aggregate (*Farran et al., 2013*). These genes were detected in all isolates however, the biofilm production ability varies from weak to strong were observed in our previous study (*Naorem et al., 2020*). Our previous study identified that SA G6 isolate obtained from skin infection showed a weak biofilm-forming ability (*Naorem et al., 2020*). The low biofilm formation in SA G6 might be degraded the biofilm by DNase enzyme found in skin cells (*Eckhart et al., 2007*). The previous study revealed that the presence of the *ica* genes did not always correlate with biofilm (*Møretrøet al., 2003*; *Nasr, Abushady & Hussein, 2012*). Some authors reported that despite the presence of *ica* operon, some staphylococcal isolates produce weak biofilm production due to the inactivation of *icaA* by insertion of IS256 (*Cho et al., 2002*; *Kiem et al., 2004*). Further reported that the insertion of IS256 inactivates *mutS* and contributes to vancomycin resistance development in vancomycin-intermediate *S. aureus* strains (*Kleinert et al., 2016*). Also, the upregulation of *icaA* and *icaD* genes during acidic stress promotes biofilm formation which in-turn plays a role to resist it from acidic and alkaline environments and establishes the niche adaptation in Staphylococcus strains (*Lindsay et al., 2006*). In addition to *ica* locus, the presence of *clfA*, *clfB*, and *epbs* genes initiates the biofilm formation (*Ghasemian et al., 2015*), however in the present study, the SA H27 isolate carried *clfA*, and *epbs* genes and showed strong biofilm formation in our previous study (*Naorem et al., 2020*) compared to other isolates while SA G8 and SA H32 isolates carried *clfA*, *clfB*, and *epbs* genes though their biofilm formation was relatively low, suggesting that presence or absence of such genes have no significant in biofilm formation. A recent study reported that *sdrC* mutant exhibited significantly inhibited biofilm formation (*Chen et al., 2019*) and the expression of the *ica* operon and *sdrC* are highly responsive to biofilm formation (*Shin et al., 2013*). Our study revealed

the sequence variation in *sdrC* in Hungarian isolates, this variation might influence the biofilm formation. The global regulatory gene, *agr* repression has been associated with biofilm formation and its induction through AIP results in seeding dispersal in mature biofilm (*Boles & Horswill, 2008*). CA-MRSA strains showed higher activity of *agr*, which controls and enhance the virulence (*Aires-De-Sousa, 2017*). It was reported that *S. aureus* strains belonged to *agr* I group exhibited a strong biofilm-forming ability than the strains belonged to *agr* IV group (*Zhang et al., 2018*) and a similar result was observed in one of our isolate SA H27. In addition to this extracellular adherence protein (encoded by *eap* gene), and beta toxin (encoded by *hlb* gene) play a role in biofilm maturation (*Huseby et al., 2010*; *Sugimoto et al., 2013*). In our finding showed that *eap* gene was present only in the SA H27 isolate and this gene might be attributed to high biofilm formation. Since biofilm formation involves many factors/ genes that take part in PIA dependent or independent biofilm, biofilm formation by regulator genes and eDNA (*Archer et al., 2011*). Also, the presence of such genes in *S. aureus* may not provide much impact on biofilm formation profiling. There was a difference in the prevalence of biofilm-associated genes between the isolated strains and suggests that the presence of genes encoding biofilm formation is not an absolute determinant of biofilm formation ability observed in our previous study (*Naorem et al., 2020*). Thus, our future studies will focus on the expression profiling of such relevant genes which may be necessary to determine the key genes involved in biofilm formation.

The high survival rates were observed in both acidic and alkaline pH conditions in all isolates was evaluated by the genomic study, it is elucidated that all the isolates possessed the arginine deiminase and urease operon that aids in the generation of ammonia due to the hydrolysis of L-arginine and urea by arginine deiminase and urease. The released ammonia and urea counteract the acidic environment (*Cotter & Hill, 2003*; *Valenzuela, Cerda & Toledo, 2003*). Further, the proton efflux pump ($F_0F_1$ ATPase) plays a role to extrude $H^+$ out of the cells and maintains the pH homeostasis (*Foster, 2004*; *Maurer et al., 2005*). However, in the case of alkaline tolerance, it was reported that the *S. aureus* genome encodes a unique Ktr-like system where the cytoplasmic gating protein KtrC regulates the uptake of $K^+$ that is essential for maintaining cytoplasmic pH and supporting $H^+$ efflux under alkaline conditions (*Gries et al., 2016*).

The ability of *S. aureus* as a pathogen can be accredited to its arsenal of virulence factors among which secreted pore-forming toxins (PFTs), exfoliative toxins (ETs), ESAT-6-like proteins, exoenzymes, and superantigens (SAgs) play a significant role in the pathogenesis of invading infections in healthy individuals (*Otto, 2014*; *Bartlett & Hulten, 2010*). The presence of *hlb* gene in the isolates contributes to the phagosomal escape of *S. aureus* and influences biofilm development (*Huseby et al., 2010*; *Periasamy et al., 2012*). The PVL toxin was identified in the prophages of Hungarian isolates and expressing Sa2 integrase. These isolates have cytolytic activity against blood cells and leukocytes, contributing to the *S. aureus* pathogenicity (*Vandenesch et al., 2003*). Staphylococcal enterotoxins (SEs) or staphylococcal superantigens proteins (SAgs) are well-known for causing food poisoning, localized epidermal infections (bullous impetigo), and generalized diseases (Staphylococcal scalded skin syndrome) (*Grumann, Nübel & Bröker, 2014*; *Argudín, Mendoza & Rodicio,*

*2010*). SEs encoding genes are located on mobile elements including bacteriophages, pathogenicity islands (SaPI), or plasmids. In this study, SEs encoding genes such as *sea, seg, sei, yent1, yent2, selk, selm, seln, and selo* were identified. Hungarian isolates, SA H27, and SA H32 acquired *seg* and *sei* genes, however *sei* gene was absent in Germany isolates, SA G6, and SA G8. These *seg* and *sei* genes belong to *egc* (enterotoxin gene cluster), involve in staphylococcal food poisoning TSS, and SSF (*Jarraud et al., 2001*; *Chen, Chiou & Tsen, 2004*) and *egc* was distributed widely in clinical isolates and playing a role in pathogenesis (*Jarraud et al., 2001*). Exfoliative toxins (ETs) are known as epidermolytic toxins that induce skin shedding and blister formation (*Melish & Glasgow, 1970*). In this study, *eta* gene encoded for ETA toxin was found in all the isolates and responsible for causing human skin damage, and most prevalent in Europe (*Ladhani, 2001*). Capsular polysaccharide synthesis genes are almost all detected in clinical isolates *S. aureus* showing significant virulence by targeting the antibodies that protect against Staphylococcal infections Suetal1997. Type VII secretion system (T7SS) was present in Germany isolates (Fig. 3A) and promoting them to persist in their hosts (*Tchoupa et al., 2019*). The *esxA* and *esxB* gene show a significant role in the distribution and colonization of *S. aureus*, and activation of the cell-mediated immune responses, boost the pathogenesis (*Burts et al., 2005*). Also, *esaD* gene found only in Germany isolates suggesting that this gene can inhibit the growth of other closely related *S. aureus* strains and playing a role in an intra-species competition (*Cao et al., 2016*). The family of beta-hemolysin converting phage encodes proteins such as SCIN (staphylococcal complement inhibitor) and CHIPS (chemotaxis inhibiting protein of staphylococcus) involved in host-pathogen interaction and contribute to evading human innate immune response (*Wamel et al., 2006*), these proteins were identified in intact prophages of SA G8 and SA G27 genomes but CHIPs was absent in the prophages of SA H32 genome. Therefore, prophages were the reservoir of virulence and resistance factors that play a role in the evolution of virulence strains and causing a major threat to human and animal health (*Barrera-Rivas et al., 2017*). The presence of ARGs and VFGs in the prophage regions of SA G6 genome differentiates it from the other *S. aureus* isolates and may determine its greater pathogenic potential by modifying its antigenicity (*Barrera-Rivas et al., 2017*). Also, plasmid p1G6 carried *qacA* gene, which is known to decrease chlorhexidine (antiseptic) susceptibility and giving an event of MGEs transfer evidence of *qacA* across the *S. aureus* strains (*Labreck et al., 2018*). The harbor of MGEs (mosaic features of prophages and plasmids) contributes to the tremendous distribution of ARGs and VFGs among the *S. aureus* isolates (*McCarthy & Lindsay, 2012*; *McCarthy et al., 2014*). This MGEs transfer event could be useful for the survival of *S. aureus* in different ecological niches (*Lindsay, 2010*).

The pangenome described here is composed of 3415 genes, of these, 1762 genes are shared among *S. aureus* isolates (Fig. 6B). Functional annotation of the core-genome revelated that they are mostly associated transcription and translation, and different metabolism categories, such similar result was reported previously (*Bosi et al., 2016*; *Sharma et al., 2018*). The core-genome and accessory genome functional characterizations revealed that *S. aureus* isolates required amino acids than carbohydrates as the energy source and suggests that these isolates adapted to grow in a protein-rich medium than carbohydrates

(Figs. 5A and 5B). It was suggested that the survival of *S. aureus* can be maintained by the catabolism of amino acids (*Halsey et al., 2017*). The core-genome has 51.6% of genes and validated that *S. aureus* is a clonal species (*Feil et al., 2003*; *Bosi et al., 2016*). The mutation event that occurred in the core-genome of closely related *S. aureus* provides important roles in virulence and persistence of *S. aureus* strains (*Kennedy et al., 2008*). So, an in-depth analysis of strain-specific genetic variation is required for further understanding of the pathogenicity. The inflation of pan-genome and deflation of core-genome was seen after the introduction of reference genomes and its regression analysis revealed that the pan-genome is open, suggesting that the gene repertoire of this species is theoretically limitless. A similar finding was observed in the DNA microarray experiment of thirty-six *S. aureus* isolates (*Fitzgerald et al., 2001*). The drastic decline of the core/pan- genome ratio after the introduction HE579071.1 (*S. aureus subsp. aureus* ST228) and SA G6 suggested that these two strains have distinct genomic contents (Fig. 6C). The genomic content variation between the genomes is due to the acquisition of certain genes that encode for virulence and resistance factors, pathogenicity islands, prophage-like elements, plasmids, mobile element proteins, and hypothetical proteins in the GIs. These GIs are mobilized across organisms via HGT events (*Schmidt & Hensel, 2004*). This finding was supported by gaps that appeared in the genome ring of SA G6 genome and suggesting that this isolate showed a distant relationship to others (Fig. 4). The gaps that appeared in the map are due to the GC% content difference in the comparative genomes, and it results from the event of MGEs transfer via HGT and the GC skewed regions indicated the regions where HGT occurred (*Hayek, 2013*).

We specifically analyzed the presence of ARGs and VFGs in the core genomes and pangenomes. Some genes involved in multidrug resistance or drug efflux such *ygaD*, *arlR*, *arlS*, and *mepA* are components of the core-genome (Fig. 2). The large repertoire of genes (29%) in the accessory genome gives advantages in adaptation and that can contribute to pathogenicity or niche specificity of strains (*Medini et al., 2005*). The analysis of pangenome is essential to understand the event of MGEs transfer and *S. aureus* evolution (*Ozer, 2018*). The interpretation from the dispersible and singleton genes content analysis of *S. aureus* genomes allows us to understand the genetic variation among the CC5 and CC22. Juhas et al. reported that most dispensable and singleton genes were acquired through HGT and operate an important role in drug resistance or virulence (*Juhas, Eberl & Church, 2012*). A high portion of unique genes or singletons in *S. aureus* genomes were related to MGEs, which could drive the gaining of novel functional elements especially drug resistance and virulence. These singletons are the main drivers of the phenotypic variation within *S. aureus* strains and the evolution of *S. aureus* (*Carvalho et al., 2019*).

The phylogenetic trees based on whole-genome and core-genome SNP methods support each other and revealed that these methods were able to distinguish between strains at a higher resolution in terms of the geographic origin of strains and phylogenetic trees are illustrated in Fig. 7. The phylogenomic analysis revealed that the strains with ST225 (Germany), ST228 (Germany, Switzerland), ST105 (USA), and ST5 (Japan) were clustered in the same CC5 clade (Cluster C), and a different clade (Cluster A) was noticed among the UK origin ST22 (CC22) and diverged from Germany origin strains (Fig. 7A), this

finding was in good agreement with the previously published article (*Aanensen et al., 2016*). The CC5 (ST225) and CC22 (ST22) were found to be the most dominant clones circulating in Europe (*Grundmann et al., 2014*). The comparative genome analysis revealed that Germany and Hungarian isolates are genetically diverse and showing variation among them due to the gain or loss of MGEs such as *SCCmec*, plasmid, phage elements, or the insertion of transposase. The event of MGEs transfer was observed in ST5, ST225, and ST228 (Fig. 7A), and similar results were also reported previously (*Nubel et al., 2008*; *Nübel et al., 2010*; *Vogel et al., 2012*). The SNPs located in the core-genome define as the element present in *S. aureus* strains, these SNPs based phylogenetic tree was constructed to avoid the HGT of MGEs misuse phylogenetic interpretation, as well as this tree, resolved the subdivision within cluster C of Fig. 7A indicating that SA G6 isolates and *S. aureus subsp. aureus* ST228 exhibits the closest strains (Fig. 7B). These strains shared the genetic background (ST228/*SCCmec*-I) and revealing 99.8% OrthoANIu similarity value in their genomes, likewise, Hungarian isolates (SA H27 and SA H32) in clade A (Fig. 7B) shared molecular epidemiological background in terms of *SCCmec*-IVa, and ST-22 and showing 99.8% OrthoANIu value. However, SA G8 isolate and *S. aureus subsp. aureus* ST228 belongs to ST225 and ST105, respectively were clustered together (Fig. 7B). The strains with the same genetic background were clustered together in both phylogenetic trees which suggest that these strains are highly alike, however comparative genome analysis exposed that the acquisition of phage elements and plasmids through the events of MGEs transfer contribute to differences in their phenotypic characters. Such events provide an impact on the fitness or pathogenicity or epidemicity of the strains.

## CONCLUSIONS

Using WGS, we characterized the four clinical MRSA isolates that infect the skin, nostrils, trachea, and other sites. The data generated from the WGS confirmed the diversity of MRSA among the same CC5 and CC22. It is clearly stated that the biofilm-forming ability of MRSA was not correlated with the presence of biofilm-forming encoding genes, also the genetic constituents have no information regarding the infection sites. So, expression profiling of biofilm-related genes is required to define the key genes involved in biofilm formation. The comparative genome study allowed the segregation of isolates of geographical origin, and differentiation of clinical isolates from the commensal isolates. An interesting finding is the addition of SA G6 genome responsible for open pan-genome and diversity among genomes. The openness of pan-genomes of *S. aureus* isolates relies on the acquisition of MGEs. The evidence of MGEs transfer event especially in SA G6 is supported by the drastic drop of the core/pan-genome ratio curve, and gaps and GC skewed regions in comparative genome map. The presence of *ant(6)-Ia, aph(3′)-III* and *sat-4* in the GI region of SA G6 are likely acquired and these genes may provide fitness and a selective advantage during host-adaptation and colonization. Phylogenetic analysis suggests that SA G6 and *S. aureus subsp. aureus* ST228 strains are distinct from its group. The acquisition of plasmid and prophage functional modules such as ARGs and VFGs in *S. aureus* isolates contributes a major role in the rapid evolution of pathogenic *S. aureus* lineages and that confer specific

advantages in a defined host under environmental conditions. Through this comparative genome analysis would improve the knowledge about the pathogenic *S. aureus* strain's characterization, adaptation, and dynamic evolutionary process in the transmission of infections globally.

## ACKNOWLEDGEMENTS

We acknowledged Dr. Schneider Gyorgy, Department of Medical Microbiology and Immunology, Medical School, University of Pecs for providing the strains. Further, we acknowledged Peter Urban for technical laboratory assistance and Gunajit Goswami for English language review. The authors acknowledge the anonymous reviewers for their valuable suggestions that helped improve the quality of the manuscript.

### Funding

This work was supported by the European Regional Development Fund (GINOP-2.3.2-15-2016-00021) and the European Regional Development Fund (GINOP-2.3.2-15-2016-00047). The Open Access of this article has been supported by the European Social Fund Grant no.: EFOP-3.6.1.-16-2016-00004 entitled by Comprehensive Development for Implementing Smart Specialization Strategies at the University of Pécs. The funders had no role in study design, data collection and analysis, decision to publish, or preparation of the manuscript.

### Grant Disclosures

The following grant information was disclosed by the authors:
European Regional Development Fund: GINOP-2.3.2-15-2016-00021.
European Regional Development Fund: GINOP-2.3.2-15-2016-00047.
European Social Fund: EFOP-3.6.1.-16-2016-00004.
Comprehensive Development for Implementing Smart Specialization Strategies at the University of Pécs.

### Competing Interests

The authors declare there are no competing interests.

### Author Contributions

- Romen Singh Naorem conceived and designed the experiments, performed the experiments, analyzed the data, prepared figures and/or tables, authored or reviewed drafts of the paper, and approved the final draft.
- Jochen Blom analyzed the data, prepared figures and/or tables, and approved the final draft.
- Csaba Fekete conceived and designed the experiments, authored or reviewed drafts of the paper, and approved the final draft.

## Data Availability

Data are available at the NCBI database under the accession numbers: RAHA00000000, QZFC00000000, CP032161, RAHP00000000.

## Supplemental Information

Supplemental information for this article can be found online at http://dx.doi.org/10.7717/peerj.10185#supplemental-information.

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
