# Peer review of "Genome-wide comparison of four MRSA clinical isolates from Germany and Hungary"

_PeerJ, doi:10.7717/peerj.10185_

## Round 0.1 · original submission · Major Revisions

Thank you for considering the Peer J for your manuscript submission titled “Whole-genome sequence and genome-wide comparison of clinical isolates methicillin-resistant Staphylococcus aureus”.

Take into account the comments of the two reviewers, I coincide with their opinions. While the referees do acknowledge that the manuscript has merit, the clear consensus is that substantial modifications are needed to substantiate the claims made. A number of topics were raised by the reviewers that should be addressed to improve the technical quality of your manuscript prior to submission of a revised version of your MS.
Therefore, I consider that the manuscript can be re-submitted as a revised version provided that the recommendations are addressed, all substantial corrections and additional experimentation suggested, which would reinforce the biological phenomenon studied. Also, it is necessary to point out the novelty of the study, in the introduction as well as in the discussion section.

Thank you for your progress, and we look forward to your resubmission.

·

Basic reporting

The manuscript requires a thorough revision of English language; it is recommended that the final English edition be done by a native English speaker or a professional language editing service. Also, there are some typographical mistakes in the body of the manuscript and in tables and figures.

Several articles are available in databases that already analyze comparative genomics of Hospital Acquired- and Community-Acquired MRSA isolates, which seems to be the case of this report. The isolates selected for this work seem to fit with these features. None of those articles are cited in this work.

There is not a clear hypothesis stated or suggested by the authors. It seems a comparative genomics study of four randomly selected Staphylococcus aureus isolates. It is important to define which were the reasons or criteria to select those clones, in order to also establish a proper hypothesis.

Experimental design

The research presented fits with the Aims and Scope of the Journal.

The main question to be resolved by the research, and the hypothesis so on, are not clearly stated. It is not clear which were the criteria to select the clones analyzed, so it is not possible to generate a research question justifying the comparative genomics.

Comparative genomics is performed with enough rigor to support the results.

Methods are described with enough detail to be reproduced and are adequate to reach the goal of comparative genomics

Validity of the findings

Since the main goal of the research is to find differences among genomes of the four S. aureus isolates tested, the manuscript is mainly descriptive of the findings and differences, with little opportunity to establish correlations with pathology, genetic background or epidemiological data related to the isolates.

Genomic analysis sounds robust and well analyzed with appropriate analysis tools.

Since there is not a clear research question or a hypothesis, little or no conclusions may emerge from the extensive description of similarities and differences of genomes.

Additional comments

The manuscript presents a detailed description of the comparative genomics of four isolates of Staphylococcus aureus resistant to methicillin (MRSA) from Germany and Hungary. Although the methodology and results are well described, authors should consider several issues to improve their manuscript before it could be accepted for publication.
The title must represent the major findings or major research topics of the manuscript and be specific. For example “Genome-wide comparison of four MRSA clinical isolates from Germany and Hungary” (obviously, to make comparative genomics, whole-genome sequencing is needed; it is not necessary to indicate it in the title).
A hypothesis is not clearly stated, and that is because there is not a clear question behind the analysis. The authors should provide a clear justification for the selection of the four isolates to analyze. Why compare Germanic with Hungarian isolates? Which is the relationship between these countries with respect to the research? Was any other criterion for the selection of the isolates (particular pathologies, representatives of an outbreak, any physiological feature that makes them interesting for research)? This is important because the rationale behind the selection of the strains may direct the search of similarities and differences and establish correlations with particular features.
Authors should also emphasize which is the particular contribution of their research to the scientific community. There are hundreds of papers already published in comparative genomics of Staphylococcus aureus strains, must of them comparing few reference, well-characterized strains, or dozens of clinical isolates.
Genomic typing of bacterial isolates of clinical importance has been intended with a variety of methods. The authors should present in the introduction a brief analysis of the advantages and disadvantages of other methods (MLST, PFGE, VNTRs, spa-typing, SCCmec-typing, agr-typing) used for Staphylococcus aureus against whole-genome sequence analysis.
Biofilm is analyzed but not presented in the introduction section as a goal of the research.
Results on biofilm formation and tolerance to pH are not properly referred in Table S1.
Improve the presentation of figure S1 by specifying the name of the strains in the lanes and the identification of the amplification products, either by size or name. Make this match with the figure caption.
Avoid superlative subjective adjectives as “intriguingly, surprisingly, unexpectedly” and attach to the quantitative or qualitative description of data.
Line 259: specify which is the antiseptic agent, name, or category.
Although low similarity at the sequence level between plasmids p1G6 and pTW20_1, it is quite evident the similarity in physical distribution and content of the genes. A more detailed description and comparative analysis are necessary.
Lines 270-272. The authors should briefly describe the distribution of toxin among strains.
Lines 291-293. If genes including different autoinducing peptides were detected among the isolates, this suggests the presence of an accessory global regulator locus (agr) which encodes the autoinducing peptides. Agr locus information is also used for agr-typing, which may be determined from the sequence and included as another molecular typing method in table 1.
Lines 295-299. There is confusion in the numbers indicating proportions, which represents the number in the denominator.
Figure 4 caption should refer to the genome accession numbers with the isolates.
In order to improve discussion, authors must take advantage of the information provided by the deduced sequence typing methods, such as MLST, spa-typing, SCCmec-typing and agr-typing (not included in this analysis). Together these methods may provide a molecular epidemiology background of these isolates in relation to comparative genomics. For example, German isolates are STs included in the Clonal Complex 5 which is typical of Hospital Acquired MRSA, while Hungarian isolates also pertain to Clonal Complex 5, but ST22 background suggests its relationship with Community Acquired MRSA. The presence of some virulence genes are particularly frequent in each kind of pathology. For example, PVL toxin was commonly used as a marker of Community Acquired MRSA and is related to particular pathologies.
Discussion is more a review of literature rather than contrasting the findings of this research with previously published papers. As I commented previously, there are no literature references to other comparative genomic studies for the discussion.
Because of the lack of a solid scientific question, conclusions presented are to general that may describe any other genomic comparison between any kind of bacterial isolates.

Reviewer 2 ·

Basic reporting

In this work Feteke et. al. analyzed and compare the genome of four methicillin-resistant Staphylococcus aureus (MRSA). The aim and scope of the article is very interesting and relevant. However, the manuscript is not clear and unambiguous, and the writing must be improved.

The authors provide enough references and field background; however, the authors could take into account the next references:

Aanensen, D. M., Feil, E. J., Holden, M. T., Dordel, J., Yeats, C. A., Fedosejev, A., ... & Chlebowicz, M. A. (2016). Whole-genome sequencing for routine pathogen surveillance in public health: a population snapshot of invasive Staphylococcus aureus in Europe. MBio, 7(3).

Kleinert, F., Kallies, R., Zweynert, A., & Bierbaum, G. (2016). Draft genome sequences of three northern German epidemic Staphylococcus aureus (ST247) strains containing multiple copies of IS256. Genome announcements, 4(5).

The structure of the article has a standard format. The figures are relevant and help to understand the paper.

MRSA genome number accession are reported.

Experimental design

Bioinformatics and experimental findings are original and within Aims and Scope of the journal. The genome analysis of MRSA is well conducted, following high technical standard. However, some experimental data is not enough to support all the findings.

Validity of the findings

Most of the results and findings are well supported. However, there are some issues that must be clarified.

1. SA H27 biofilm formation was almost four times bigger than other three strains, however, this finding did not were described neither discussed.
2. Since involved genes in biofilm formation are present in all four strains, an expression gene/protein experiment must be designed and performed to understand this point.
3. Strain SA G6 is proposed to be more pathogenic than other three strain because of HGT gained genes; this is speculative, and an experimental approach must be carried out to prove this point.
4. To validate HGT of SA G6 genes, a phylogenetic analysis must be performed.
5. In the phylogenetic analysis (Figure 7C) it will be desirable to increase the number of OTUs to gain knowledge about strains origin.
6. Phylogenetic analysis was poorly discussed; this section must be re-written.
7. There was no discussion about the findings about SA pan-genome.

Minor corrections
• Homogenize name plasmids in text line 258 and 260 within table 1.
• Correct pan-genome in line 428.
• Correct syntaxis in line 519 (but sei gene was not found in Germany isolates).
• Correct SA H27 and SA G8 (separate acronyms) in line 539.
• All scientific names should be italicized or underlined in figure legends.
• In figure 4, strain names, SA G6, SA G8, SA H27, SA H32, must be showed to identification of strains.
• Figure 7A and 16S rRNA analysis must be deleted since this tree has to low information to resolve taxonomy at strain level.

---

## Round 0.2 · Minor Revisions

The authors must take into account the corrections and comments made by the reviewers, especially the one referring to the English language edition. Although it will not be possible for the authors to carry out additional assays, they should evaluate the proposals of the reviewer 2, or eliminate conclusions based on speculations.

·

Basic reporting

The manuscript has been revised for the English style and is now more properly written. However, there are still some details in verb conjugation along the manuscript, please review grammar in more detail. Again, a professional language editing service may be useful. Authors have reviewed literature on comparative genomics of MRSA genomes and included it for introduction and discussion. Authors have also inserted an hypothesis and justification for the selection of the strains, so the comparative genomics is now justified.
Literature references were complemented and are in accordance with the results analysis and discussion.
Figures are relevant for the manuscript.

Experimental design

The research presented fits with the Aims and Scope of the Journal.
The main question has now been stated and an hypothesis has been presented, so the comparative genomics analyisis is justified.
Comparative genomics is performed with enough rigor to support the results.
Methods are described with enough detail to be reproduced and are adequate to reach the goal of comparative genomics

Validity of the findings

Differences in the genetic background of the selected strains were already known by the authors and now they have been provided to enrich the genomic analysis and the interpretation of the findings.
Genomic analysis is robust and well analysed with appropriate analysis tools.
Clear genetic background description of the tested strains is now available, and certainly helps to clarify relatedness and differences among strains.

Additional comments

The manuscript presents a detailed description of the comparative genomics of four isolates of Staphylococcus aureus resistant to methicillin (MRSA) from Germany and Hungary. Although the methodology and results are well described, authors should consider several issues to improve their manuscript before it could be accepted for publication.
Abstract: Although a hypothesis has been elaborated, this must be clearly stated in the abstract, and clear identification of the strains tested in the work must be included. The abstract content is clumpsy but it contains the information that is relevant to the research.
All of my previous suggestions on information presentation, analysis and discussion were attended, and were properly directed to improve the contributions of the manuscript to general knowledge on the topic. Besides, they included an important analysis of gene distribution in pangenome, core genome and distribution of singletons among strains that contribute to understand potential pathogenic features of the analysed strains.
Some minor observations.:
Line 34: indicate the identifier of the Hungarian isolate
Line 111. An example of the need of reviewing grammar; authors cite a previously published work and conjugate “includes” in present tense.
Lines 251 to 253. There is no reference to the table that present those results.
Lines 255 to 262. This phrase is to long; divide in several phrases and improve syntaxis.
Line 265: The term “epidemiologic characteristics” refer to the surrounding, environmental or social factors that are related to the outcome of a disease. The authors refer to molecular techniques that are frequently used in molecular epidemiology studies that provide information on the genetic background of the strains. Please rephrase to fit these concepts.
Line 285. Probably authors mean “phage transduction”.
Line 306. Table S1. Please correct the Greek letter in beta-lactamase.
Line 502. Correct the Greek letter in beta-lactamase
Line 638. “Distribution” instead of “diversity”?
Lines 685 to 687. It is not clear to me why this conclusion: “….thus HGT was not limited within S. aureus strains”. Does it mean that authors have evidence of HGT from other species or from strains outside of this study? Which is that evidence? Please clarify.

Reviewer 2 ·

Basic reporting

In this work Feteke et. al. performs a genome analysis of four strains from methicillin-resistant Staphylococcus aureus (MRSA). The aim and scope of the article is very interesting and relevant. However, the manuscript is not clear and unambiguous, and the writing must be improved since it is difficult to follow in some parts. Some examples are:

• It is not clear the origin and reasons why the four strains were selected from the previous study (lines 110-124).
• MGE abbreviation was not defined (line 42).
• Results from pH tolerance assay were not referenced to the corresponding table.
• There are incomplete ideas (e.g. line 272-274).
• Ideas from lines 377-383, 439-442 were not understandable.
• There is a discordance between text and figure 6A SA G6 singletons number (143 vs 220).

An all-text detailed revision should be done.

Experimental design

Bioinformatics and experimental findings are original and within Aims and Scope of the journal. The genome analysis of MRSA is well conducted, following high technical standard.

However, some experimental data is not enough to support all the findings.

Validity of the findings

Most of the results and findings are well supported.

However, there are some issues that must be clarified.

1. Since involved genes in acidic and alkaline pH conditions resistance are present in all four strains, an expression gene experiment must be designed and performed to validate this point.
2. It was not clear the reason to use S. aureus subsp. aureus HO 5096 0412 to compare al genomes in figure 4.
3. A correlation between ARGs identification and antibiotic resistance analysis is mentioned (line 322), but antibiotic resistance analyses were not done in this study.
4. Biofilm formation was widely discussed; however, biofilm formation analyses were not present in this work.
5. It was discussed that S. aureus subsp. aureus ST228 (HE579071.1) and SA G6 addition to pangenome drastic decline of the core/pan- genome ratio, however, ¿in four strain pangenome analysis this decline was observed when SA G6 was added?
6. To validate that singletons genes could be acquired by HGT (line 644-646), a datailed bioinformatic analysis must be performed.
7. In the phylogenomic analysis, ST105, ST22 and ST5 are central to understand this result, however it is not clear its origin and relevance.

---

## Round 0.3 · accepted · Accept

Once the authors fulfilled most of the requirements made by the reviewers, which contributed positively to improve the understanding of the study, as well as highlighting its importance in the field of study. I am pleased to inform you that it has met the scope of the editorial to be accepted and published in PeerJ.